# Efficacy of Hepatitis B Virus Vaccines HBVaxpro40© and Fendrix© in Patients with Chronic Liver Disease in Clinical Practice

**DOI:** 10.3390/vaccines10081323

**Published:** 2022-08-16

**Authors:** Diana Horta, Montserrat Forné, Anna Agustí, Agnes Raga, Albert Martín-Cardona, Juana María Hernández-Soto, Pablo Ruiz-Ramírez, Maria Esteve-Comas

**Affiliations:** 1Department of Gastroenterology and Hepatology, Hospital Universitari Mútua Terrassa, Universitat de Barcelona, 08221 Terrassa, Spain; 2Centro de Investigación Biomédica en Red de Enfermedades Hepáticas y Digestivas (CIBERehd), Instituto de Salud Carlos III, 28029 Madrid, Spain

**Keywords:** vaccination, hepatitis B vaccination, hepatitis B, chronic liver disease, liver cirrhosis

## Abstract

Chronic liver disease results in a low response rate to the hepatitis B virus vaccine. Information on the efficacy of the double adjuvanted vaccine FENDRIX^®^ (3-O-desacyl-4’-monophosphoryl lipid A and aluminum phosphate) and single adjuvant HBVAXPRO^®^40 (aluminum hydroxyphosphate sulfate) in chronic liver disease is scarce. The primary aim of this prospective study in clinical practice was to evaluate the effectiveness of HBVAXPRO^®^40 and FENDRIX^®^ in this setting. Patients received HBVAXPRO^®^ (0, 1 and 6 months) or FENDRIX^®^ (0, 1, 2 and 6 months) depending on availability. Clinical data and anti-HBs levels were collected at 2, 6 and 12 months. A total of 125 patients were included (mean age 61.8 years; 57.6% males; 43.2% liver cirrhosis; 75.9% Child A and 24.1% Child B): 76 were vaccinated with HBVAXPRO^®^ and 49 with FENDRIX^®^. There were no significant differences between the two vaccines. The overall response rates at 2, 6 and 12 months were 76.8, 72.8 and 59.2%, respectively. In the univariate analysis, active alcohol intake, alcohol etiology, liver cirrhosis and ultrasound signs of portal hypertension were associated with a lower response to vaccination, whereas in the multivariate analysis, liver cirrhosis was the only factor that significantly increased the likelihood of nonresponse (OR 10.5). HBVAXPRO^®^ and FENDRIX^®^ are good options for HBV vaccination in patients with chronic liver disease.

## 1. Introduction

Cirrhotic patients present innate and adaptive immune dysfunction, which results in an increased risk of contracting infections [1]. In these patients, it is important to prevent hepatitis B virus (HBV) infection, as it can lead to impaired liver function and/or complications [2]. Therefore, the administration of HBV and hepatitis A virus (HAV) vaccines is recommended to all patients with chronic liver disease to whom infection by these viruses may present an unfavorable course [3,4,5].

However, the efficacy of both vaccines in this group of patients is lower than that in the healthy population, especially in advanced liver disease.

The HBV vaccine has been shown to be effective in the immunocompetent adult population (patients under 40 years of age) with a 95% seroconversion rate [6,7]. The recommended dose is 20 µg at 0, 1 and 6 months.

Seroprotection was defined as having hepatitis B surface antibody (anti-HBs) titers equal to or greater than 10 U/mL [8]. The protection offered by primary HBV vaccination with a monovalent vaccine to healthy patients when immunized in childhood can persist throughout their lifetime despite the progressive loss of circulating antibody levels below 10 U/mL [7].

There are several factors associated with a lower response to vaccination, such as smoking, male sex, obesity and age over 40 years [9,10,11].

In addition, the effectiveness of HBV vaccination has been shown to be lower in patients with chronic renal failure on dialysis [12], in patients with human immunodeficiency virus (HIV) infection [13] and in patients with inflammatory bowel disease [14] (considering that the response worsens with age). For this reason, some studies have been designed to demonstrate greater efficacy of double-dose HBV vaccination in this subgroup of patients [12,13,14].

Chronic liver disease also results in a lower response to vaccination. Seroprotection at standard doses is low (16–20%) [15,16,17]. Despite increasing standard doses to double doses or administering an accelerated regimen, the HBV vaccine response rate increases but remains suboptimal [9,10,18,19].

The FENDRIX^®^ vaccine (GlaxoSmithKline, Rixensart, Belgium) is a double adjuvant HBV vaccine containing the adjuvant 3-O-desacyl-4’-monophosphoryl lipid A and aluminium phosphate. Although the cost of this vaccine is higher than that of the nonadjuvanted vaccine (ENGERIX B^®^, GlaxoSmithKline, Rixensart, Belguim), this adjuvanted vaccine could induce higher immunity in patients with chronic liver disease. In a randomized experimental study with 93 liver transplant candidates, the safety and immunogenicity of two doses of FENDRIX^®^ vs. three double doses of ENGERIX B^®^ (GlaxoSmithKline, Rixensart, Belguim) showed that the percentage of patients with seroprotective anti-HBs levels was significantly higher in the FENDRIX^®^ (GlaxoSmithKline, Rixensart, Belguim) group (60%) versus the Engerix B group (32%) [20]. On the other hand, HBVAXPRO^®^ 40 (Sanofi Pasteur MSD, Lyon, France) is a single adjuvant HBV vaccine with aluminium hydroxyphosphate sulfate, approved for predialysis, peritoneal dialysis and hemodialysis patients aged ≥18 years.

There are few studies evaluating the effectiveness of HBV vaccination in patients with chronic liver disease in clinical practice [15,16,17,19,21,22]. Most studies are retrospective with small samples or only include liver transplant candidates with heterogeneous results. In addition, they do not represent the full spectrum of chronic liver disease, and few include all the etiologies of chronic liver disease [2,10,23]. In addition, there are no studies evaluating the efficacy of FENDRIX^®^ (GlaxoSmithKline, Rixensart, Belguim) in patients with chronic liver disease in clinical practice.

Thus, we aimed to (1) evaluate the effectiveness of the HBV vaccines HBVAXPRO^®^ 40 (Sanofi Pasteur MSD, Lyon, France) and FENDRIX^®^ (GlaxoSmithKline, Rixensart, Belguim) in patients with chronic liver disease in a real-life setting, (2) assess the effectiveness of both vaccines at 6 and 12 months and (3) assess the occurrence of associated HBV vaccination adverse effects (AEs).

## 2. Materials and Methods

### 2.1. Study Overview

This is a prospective observational study in clinical practice using HBVAXPRO^®^ 40 (Sanofi Pasteur MSD, Lyon, France) and FENDRIX^®^ (GlaxoSmithKline, Rixensart, Belguim) involving chronic liver disease patients with HBV-negative serologic tests. The study was conducted between September 2017 and August 2021.

### 2.2. Participants and Eligibility Criteria

Noncirrhotic and cirrhotic liver disease patients with HBV-negative serologic tests were consecutively and prospectively included in the outpatient liver unit of the Hospital Universitari Mutua Terrassa. Liver disease was assessed by abnormal liver blood tests lasting for more than 6 months, elastography, noninvasive markers of fibrosis, ultrasound and biopsy, when available. The inclusion criteria were as follows: (1) chronic liver disease patients (noncirrhotic and cirrhotic) diagnosed by liver biopsy and/or noninvasive methods (by standard clinical, analytical and ultrasound criteria); (2) HBV-negative serologic tests, including hepatitis B surface antigen (HBsAg), anti-HBs and hepatitis B core antibody (anti-HBc); and (3) 18 years old or older. The exclusion criteria were as follows: (1) allergy to vaccine components (sodium chloride, aluminum phosphate); (2) active or past HBV infection; (3) patients previously vaccinated against HBV (regardless of response); (4) Child–Pugh class C; (5) conditions that cause immunosuppression (HIV infection, chronic renal failure, active neoplasia); (6) pregnancy or breastfeeding; or (7) previous HAV infection or anti-HAV vaccination. The study was approved by the institutional review board (TASDA_003) and complied with the provisions of the Good Clinical Practice guidelines and the Declaration of Helsinki. Written informed consent was obtained from all participants, in accordance with local institutional review board requirements. Vaccination and study assessments were performed at the Digestive Unit of Hospital Universitari Mutua Terrassa. The study was registered in the Clinical Trials Registry at http://clinicaltrials.gov (NCT03459521, accessed on 9 March 2018).

### 2.3. Intervention

Serological markers of HBV infection were measured in all outpatients with noncirrhotic and cirrhotic liver disease (HBsAg, anti-HBs and anti-HBc). After informed consent was obtained, all seronegative patients were prospectively included.

Patients were vaccinated with HBVAXPRO^®^ 40 (0, 1 and 6 months) or FENDRIX^®^ (0, 1, 2 and 6 months) depending on the type of vaccine available to the Health Department of the Catalan Government.

Clinical and blood tests and the presence of anti-HBs levels were collected at 2, 6 and 12 months after the last dose of vaccination.

The HBV vaccination protocol was as follows: HBVAXPRO^®^ (Sanofi Pasteur MSD, Lyon, France) (40 µg of HBsAg) at 0, 1 and 6 months or FENDRIX^®^ (GlaxoSmithKline, Rixensart, Belguim) (20 µg of HBsAg + 50 µg of -3-O-desacyl-4’-monophosphoryl lipid) at 0, 1, 2 and 6 months. Both were administered intramuscularly in the deltoid muscle (Figure 1).

Demographic and clinical data were recorded, including age, sex, smoking habits, alcohol consumption, body mass index, the presence of other medical conditions, as well as the etiology of chronic liver disease, noninvasive serologic tests (FIB-4, APRI, FORNS) [24], liver function tests, transient elastography values, indirect ultrasound signs of portal hypertension, and treatment parameters. Finally, vaccination-related adverse events (AEs) were analyzed.

### 2.4. Definitions

A classic response to HBV vaccination was defined as anti-HBs ≥ 10 U/mL, and a nonclassical response was defined as anti-HBs ≥ 100 U/mL determined two months after the last dose of HBV vaccination [8].

### 2.5. Endpoints

The primary endpoint was the effectiveness of the HBV vaccines HBVAXPRO^®^ 40 and FENDRIX^®^ 2 months after the last dose of HBV vaccination. Secondary endpoints were (1) HBV vaccination response at 6 and 12 months; (2) AEs related to HBV vaccination; and (3) assessment of factors associated with HBV effectiveness.

### 2.6. Statistical Analysis

Quantitative data are reported as the means ± standard deviations (SD), and categorical variables are presented as frequencies and percentages. Groups were compared using the t test for quantitative variables when appropriate, and Fisher’s exact test was used for categorical variables. A multivariate logistic regression analysis was performed to evaluate variables associated with the vaccine response. The level of significance was set at two-sided 5%. All calculations and analyses were performed with the SPSS 25.0 software package (SPSS Inc.; Hp Chicago, IL, USA).

## 3. Results

### 3.1. Baseline Characteristics

A flow chart for the study is provided in Figure 2. The study included 185 outpatients with chronic liver disease, of whom 179 agreed to participate in the study. Of them, 36 had one or more exclusion criteria and were not included in the study. A total of 86 participants received HBVAXPRO^®^ 40 (Sanofi Pasteur MSD, Lyon, France) and 57 received FENDRIX^®^ (GlaxoSmithKline, Rixensart, Belguim) and 18 patients dropped out. Reasons for drop-out are shown in Figure 2. The final study cohort comprised 125 patients: 76 in the HBVAXPRO^®^ 40 (Sanofi Pasteur MSD, Lyon, France) group and 49 in the FENDRIX^®^ (GlaxoSmithKline, Rixensart, Belguim) group.

Baseline characteristics are depicted in Table 1 and were comparable between the study groups except for smoking, which was more frequent in the HBVAXPRO^®^ group (Sanofi Pasteur MSD, Lyon, France) (*p* = 0.001). The median age was 61.8 years (39–83), 57.6% (*n* = 72) were male and 43.2% (*n* = 54) had liver cirrhosis, 75.9% of whom were Child–Pugh A and 24.1% Child–Pugh B.

The etiology of chronic liver disease was 35.2% alcohol, 28.8% metabolic-associated fatty liver disease, 20% hepatitis C virus and 16% autoimmune hepatitis/primary biliary cholangitis. Regarding other medical conditions, 21.6% had diabetes, 83.3% had a BMI ≥ 25, 68% were smokers and 32% had active alcohol abuse.

### 3.2. Vaccination Efficacy

The overall response rate two months after the last dose of vaccination was 76.8% (*n* = 96) (Figure 3). The median anti-HBs level in responders after the complete vaccination was 375.5 U/mL. The HBVAXPRO^®^ 40 (Sanofi Pasteur MSD, Lyon, France) response rate at two months was 72.4% (*n* = 55) (anti-HBs ≥ 10 U/mL) and 52.6% (*n* = 40) (anti-HBs ≥ 100 U/mL), whereas the FENDRIX^®^ (GlaxoSmithKline, Rixensart, Belguim) response rate was 83.7% (*n* = 41) (anti-HBs at two months was 72.4% (*n* = 55)) (anti-HBs ≥ 10 U/mL). No significant differences between the two vaccine protocols were detected (anti-HBs *n* = 41). The geometric mean titers (GMTs) of anti-HBs with HBVAXPRO^®^ (Sanofi Pasteur MSD, Lyon, France) and FENDRIX^®^ (GlaxoSmithKline, Rixensart, Belguim) were 59.5525 ± 11.1742 U/mL and 134.0293 ± 8.9071 U/mL, respectively. No differences were observed between the two different vaccines (*p* = 0.059).

Active alcohol intake (*p* = 0.04), alcoholic etiology (*p* = 0.001), liver cirrhosis (*p* < 0.001), advanced fibrosis (F3–F4) by elastography (*p* < 0.001) and ultrasound signs of portal hypertension (*p* = 0.005) were associated with a lower response rate to vaccination in the univariate analysis (Table 2). In the multivariate analysis, only the presence of liver cirrhosis (OR 10.5; 95% CI 3.6–30.3; *p* < 0.001) was independently associated with a lower effectiveness of vaccination.

### 3.3. Vaccination Efficacy at Six and Twelve Months

At 6 months, the overall response rates were 72.8% (*n* = 91) (anti-HBs ≥ 10 U/mL) and 43.2% (*n* = 54) (anti-HBs ≥ 100 U/mL). The median anti-HBs level in responders was 286 (2–1000) U/mL. There were no significant differences between the two vaccine protocols at six months (anti-HBs ≥ 10 U/mL *p* = 0.100) (anti-HBs ≥ 100 U/mL *p* = 0.356) (Figure 4). The GMTs of anti-HBs with HBVAXPRO^®^ (Sanofi Pasteur MSD, Lyon, France) and FENDRIX^®^ (GlaxoSmithKline, Rixensart, Belguim) were 40.9826 ± 10.5988 U/mL and 81.5455 ± 8.900 U/mL, respectively. No differences were observed between the two different vaccines (*p* = 0.104).

At 12 months, the overall response rates were 59.2% (*n* = 74) (anti-HBs ≥ 10 U/mL) and 41.9% (*n* = 52) (anti-HBs ≥ 100 U/mL). The median anti-HBs levels in responders were 250.7 (2–1000) U/mL. There were no significant differences between the two vaccine protocols at twelve months (anti-HBs ≥ 10 U/mL p = 0.576) (anti-HBs ≥ 100 U/mL *p* = 0.271) (Figure 4). The GMTs of anti-HBs with HBVAXPRO^®^ (Sanofi Pasteur MSD, Lyon, France) and FENDRIX^®^ (GlaxoSmithKline, Rixensart, Belguim) were 30.0746 ± 10.5905 U/mL and 56.8721 ± 10.1744 U/mL, respectively. No differences were observed between the two different vaccines (*p* = 0.143).

Diabetes, alcoholic etiology, advanced fibrosis (F3–F4), liver cirrhosis and ultrasound signs of portal hypertension were associated with a lower response rate to vaccination at 6 months. In the multivariate analysis, liver cirrhosis (OR 5.9; 95% CI 2.4–14.3; *p* < 0.001) was independently associated with a lower effectiveness of vaccination.

At 12 months, the same clinical characteristics (except for diabetes) were found in the univariate and multivariate analyses (liver cirrhosis; OR 6.4; 95% CI 2.9–14.4; *p* < 0.001). (Table 3 and Table 4).

### 3.4. Adverse Events

There were no AEs reported during the study period.

## 4. Discussion

The majority of studies assessing HBV vaccine response have only included liver transplant candidates [9,11,15,16,17,20,21,22,25], cirrhotic patients [2,19] or patients with chronic HCV [18,26,27,28]. However, the HBV vaccine is strongly recommended in patients with chronic liver disease from all etiologies, even in the absence of liver cirrhosis. Therefore, we report herein the vaccination response rate of HVB vaccination in an outpatient liver unit attending to patients with a variety of liver diseases with different degrees of severity. In addition, two types of HBV vaccines (FENDRIX^®^, GlaxoSmithKline, Rixensart, Belguim and HBVAXPRO^®^ 40 (Sanofi Pasteur MSD, Lyon, France) with an increased immunogenic response were used depending on the availability in the setting of the vaccination program of the Health Department of the Catalan autonomous community.

We found an overall good response rate (76.8%) two months after completing the vaccination protocol with both vaccination schedules: 83.7% in the FENDRIX^®^ (GlaxoSmithKline, Rixensart, Belguim) group and 72.4% in the HBVAXPRO^®^ 40 (Sanofi Pasteur MSD, Lyon, France) group. There were also no differences in the GMT between the two vaccines, although a tendency to have higher values with the Fendrix vaccine was also observed, suggesting that it could have a better efficacy. In addition, vaccination efficacy, measured by anti-HBs levels, decreased over time (overall response rates at six and twelve months were 72.8% and 59.7%, respectively).

The HVB vaccination response rate in prior studies is heterogeneous due to the use of different vaccination regimens and the heterogeneity of the patients included. In cirrhotic patients, despite the use of double doses or accelerated regimens of standard HVB vaccines to increase efficacy, the response rate remains suboptimal (23–68%) [2,9,19,21,22]. In noncirrhotic chronic liver disease patients, HBV vaccine efficacy increases but is lower than in the healthy population [9,10,19].

To date, few studies have included patients with chronic liver disease with different etiologies [2,10,23]. Artaza et al. [2] administered three doses of 20 µg at 0, 1 and 6 months to 194 patients (107 noncirrhotic and 87 cirrhotic patients Child–Pugh A), obtaining response rates of 63.4% and 47.3%, respectively. De Maria et al. [10] included 224 subjects with chronic liver disease (138 noncirrhotic and 86 with liver cirrhosis), obtaining 62% and 42% response rates, respectively, after administering three doses of Engerix-B^®^ (GlaxoSmithKline, Rixensart, Belguim) 40 µg at 0, 1 and 2 months. Finally, Aziz et al. [23] administered three doses of 80 µg HBV vaccine to patients with chronic liver disease who had previously failed HBV vaccination and found a 52% immune response in cirrhotic patients and 83% in noncirrhotic patients. All studies, as was the case in ours, excluded decompensated cirrhotic patients.

The good vaccination response obtained in our study could be in part explained by the fact that 56.8% of the patients included were noncirrhotic patients, emphasizing the need for early adherence to a vaccination program. In contrast to previous studies, we also included Child–Pugh B patients who demonstrated a worse vaccine response in patients with complicated cirrhosis. However, direct comparison to other studies is difficult due to the use of different vaccine doses, schedules and patient characteristics.

Active alcohol intake, alcoholic etiology, liver cirrhosis and indirect signs of portal hypertension were associated with a lower response to HBV vaccination. At six months, diabetes was also associated with a lower immunization response. Several risk factors for nonresponse have been previously identified, including older age, male sex, smoking, obesity and diabetes [9,10,29]. Our data confirm some but not all of these findings from previous studies.

Alcohol consumption alters both innate and adaptive immunity and is associated with a decreased frequency of lymphocytes and an increased risk of both bacterial and viral infections [30]. Moreover, diabetes can reduce chemotaxis and neutrophil/macrophage phagocytic function, inhibiting activation of the complement cascade and cell-mediated immunity [31].

Liver cirrhosis and advanced fibrosis are well-established factors influencing a low response to HBV vaccination. As previously mentioned, this aspect has been especially evaluated in patients with end-stage liver cirrhosis awaiting liver transplant, in which accelerated or double-dose vaccination must be considered [32]. The lower response is a consequence of an impaired immune response in these patients. However, this impaired response is also observed in the earliest stages, since in our study, including only patients with Child–Pugh A and B stages, the presence of liver cirrhosis was the only factor independently related to a lower response rate.

Anti-HBs titers ≥ 10 U/mL are considered a reliable marker of protection against HBV infection. However, anti-HBs titers decline over time, as seen in the previous literature in other clinical settings [8]. In our study, we found a decrease in anti-HBs titers during follow-up, as observed in other studies performed in different epidemiological contexts [22,33]. We confirm, therefore, the need to check anti-HBs during follow-up, as the HBV vaccination response rate decreases over time. However, even in cases of loss of circulating antibodies, it has been suggested that patients maintain a certain degree of protection against HBV. In addition, we performed a second cycle or a booster dose of HBV vaccination in nonresponders or patients who presented anti-HBs loss over time. We did not include these data, as this was not the aim of the present study, but a response rate was achieved in 41.6% (5/12) and 87.5% (7/8) of nonresponders and losers of response, respectively.

There are several limitations of this study. We did not randomize patients to receive either HBVAXPRO^®^ 40 (Sanofi Pasteur MSD, Lyon, France) or FENDRIX^®^ (GlaxoSmithKline, Rixensart, Belguim) and did not have a control group. However, the study was designed in a clinical practice setting, and we did not find significant differences in either vaccine protocol. In addition, we did not include patients who were not immunized for HAV. In this group of patients, as recommended in clinical practice, we used a combined HAV/HBV vaccine [34]. Since the HAV vaccine may induce an adjuvant response to the HBV vaccine [35], these patients were not included to assess the isolated response rate to the HBV vaccine. Finally, we limited the effectiveness of the two vaccine regimens to antibody immune responses at two, six and twelve months. Future studies should include other measures of adaptive immunity, since patients with anti-HBs levels below 10 U/mL may still be protected.

## 5. Conclusions

The results of our study show that HBVAXPRO^®^ 40 (Sanofi Pasteur MSD, Lyon, France) and FENDRIX^®^ (GlaxoSmithKline, Rixensart, Belguim) are effective and safe in patients with chronic liver disease. Indications for these vaccines should be expanded to enable vaccination of chronic liver disease patients in early stages.

## Figures and Tables

**Figure 1 vaccines-10-01323-f001:**
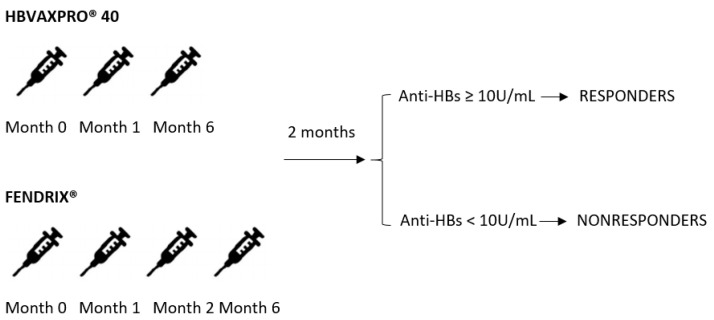
Vaccination schedule used in the study.

**Figure 2 vaccines-10-01323-f002:**
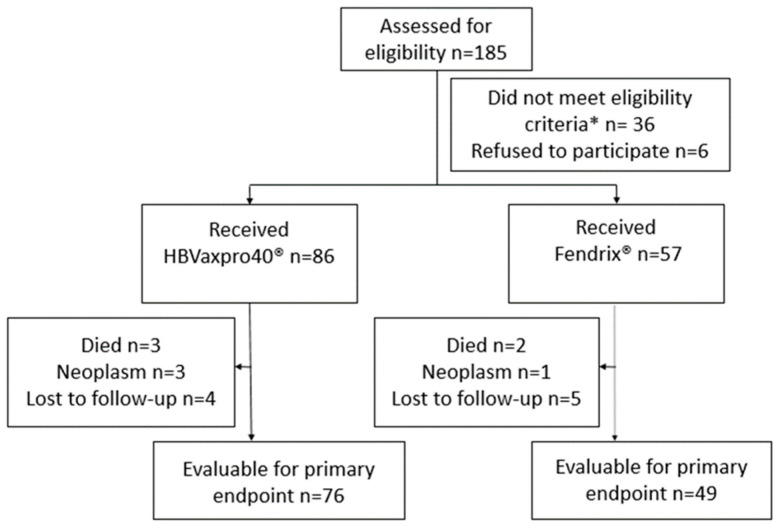
Flowchart of the study. * Patients previously vaccinated against HBV *n* = 10, Anti-HBs positive/anti-HBc negative *n* = 6, Anti-HBc positive/anti-HBs negative *n* = 9, Nonimmunized HAV infection *n* = 6, active neoplasia *n* = 2, Child–Pugh C *n* = 3.

**Figure 3 vaccines-10-01323-f003:**
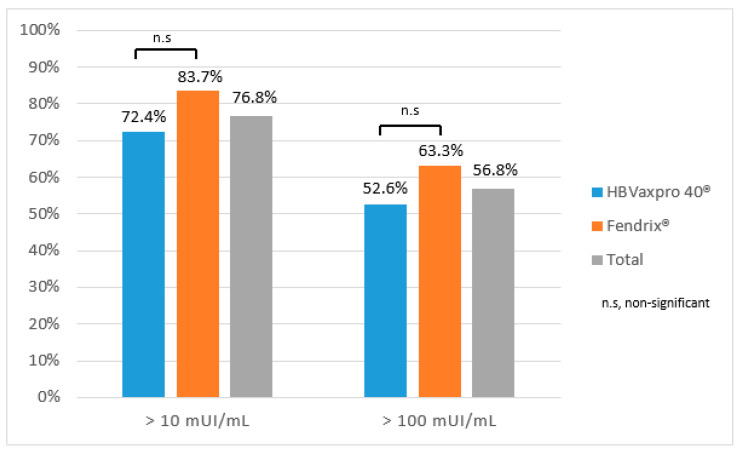
HBV vaccine response at two months. n.s, non-significant.

**Figure 4 vaccines-10-01323-f004:**
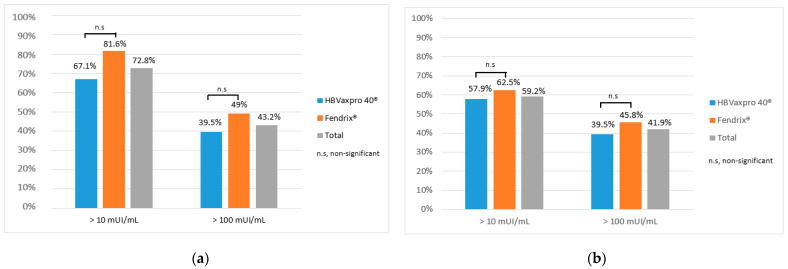
HBV vaccine response at six and twelve months. (**a**) HBV vaccine response at six months; (**b**) HBV vaccine response at twelve months. n.s, non-significant.

**Table 1 vaccines-10-01323-t001:** Baseline characteristics of patients included in the study.

	Total *n* = 125	Hbvaxpro^®^ 40 *n* = 76	Fendrix ^®^ *n* = 49	*p*
Age (Mean ± SD)	61.8 ± 9.4	62.2 ± 10.0	61.2 ± 8.5	n.s
Gender (male) (*n*, %)	72 (57.6)	42 (55.3)	30 (61.2)	n.s
Smokers (*n*, %)	85 (68)	62 (81.6)	23 (46.9)	0.001
Active alcohol intake (*n*, %)	40 (32)	20 (26.3)	20 (40.8)	n.s
Diabetes (*n*, %)	27 (21.6)	18 (23.7)	9 (18.4)	n.s
BMI (Mean ± SD)	29.5 ± 5.5	30.2 ± 5.3	28.3 ± 5.7	n.s
Overweight (BMI ≥ 25) (*n*, %)	105 (84)	69 (90.7)	36 (73.5)	n.s
Chronic liver disease				n.s
Etiology (*n*, %)				
Alcohol	44 (35.2)	25 (32.9)	19 (38.8)	
MAFLD	36 (28.8)	19 (25)	17 (34.7)	
HCV	25 (20)	18 (23.7)	7 (14.3)	
AIH/PBC	20 (16)	14 (18.4)	6 (12.2)	
Liver cirrhosis (*n*, %)	54 (43.2)	28 (36.8)	26 (53.1)	n.s
Child–Pugh (*n*, %)				n.s
A	41 (75.9)	22 (78.6)	19 (73.1)	
B	13 (24.1)	6 (21.4)	7 (26.9)	
Stage by elastography				n.s
(*n*, %)				
F0–F2	70 (56)	48 (64)	22 (45.8)	
F3–F4	52 (42.4)	27 (36)	26 (54.2)	
Index for Liver				n.s
Fibrosis (Mean ± SD)				
FIB-4	2.5 ± 2.1	2.3 ± 1.8	2.9 ± 2.7	
APRI	0.7 ± 0.8	0.6 ± 0.5	0.8 ± 1.0	
FORNS	6.4 ± 2.2	6.2 ± 2.1	6.7 ± 2.2	
US signs of portal hypertension (*n*, %)	30 (24)	14 (18.4)	16 (32.7)	n.s
Immunosuppressive treatment (*n*, %)	11 (8.8)	7 (9.2)	4 (8.2)	n.s

Abbreviations: AIH, autoimmune hepatitis; BMI, body mass index; HCV, hepatitis C; MAFLD, metabolic-associated fatty liver disease; n.s, non-significant; PBC, primary biliary cholangitis, US: ultrasound.

**Table 2 vaccines-10-01323-t002:** Comparison of patient characteristics between responders and nonresponders to HBV vaccination by univariate analysis.

	Responders *n* = 96	Nonresponders *n* = 29	OR (95% CI)	*p*
Age (years) (Mean ± SD)	61.3 ± 9.9	63.4 ± 7.5	-	n.s
Gender (male) (*n*, %)	51 (53.1)	21 (72.4)	1.36 (1.01–1.82)	n.s
Smoker (*n*, %)	64 (66.7)	21 (72.4)	1.08 (0.83–1.41)	n.s
Active alcohol intake (*n*, %)	26 (27.1)	20 (26.3)	1.78 (1.08–2.93)	0.04
Diabetes (*n*, %)	17(17.7)	10 (34.5)	1.94 (1.00–3.77)	n.s
BMI (Mean ± SD)	29.3 ± 5.6	30.1 ± 5.1	-	n.s
Overweight (BMI ≥ 25) (*n*, %)	79 (82.3)	26 (89.7)	1.08 (0.93–1.27)	n.s
Chronic liver disease				
Etiology (*n*, %)				
Alcohol	26 (27.1)	18 (62.1)	2.29 (1.48–3.53)	0.001
MAFLD	29 (30.2)	7 (24.1)	0.79 (0.39–1.63)	n.s
HCV	23 (24)	2 (6.9)	0.28 (0.07–1.14)	n.s
AIH/PBC	18 (18.8)	2 (6.9)	0.36 (0.09–1.49)	n.s
Liver cirrhosis (*n*, %)	30 (31.3)	24 (82.8)	2.64 (1.88–3.72)	<0.001
Child–Pugh (*n*, %)				
A	23(76.7)	18 (75)	0.97 (0.72–1.32)	n.s
B	7 (23.3)	6 (25)	1.07 (0.41–2.76)	n.s
Stage by elastography				
(*n*, %)				
F0–F2	64 (67.4)	6 (21.4)	0.31 (0.15–0.65)	n.s
F3–F4	31 (32.6)	22 (78.6)	2.40 (1.70–3.41)	<0.001
Index for Liver Fibrosis			-	
(Mean ± SD)				
FIB-4	2.2 ± 2.1	3.8 ± 2.3		n.s
APRI	0.6 ± 0.7	1.1 ± 0.7		n.s
FORNS	6.1 ± 2.1	7.8 ± 1.9		n.s
US signs of portal hypertension (*n*, %)	17 (17.7)	13 (44.8)	2.53 (1.40–4.57)	0.005

Abbreviations: CI: confidence interval; AIH, autoimmune hepatitis; BMI, body mass index; HCV, hepatitis C; MAFLD, metabolic-associated fatty liver disease; n.s, non-significant; PBC, primary biliary cholangitis; US: ultrasonographic.

**Table 3 vaccines-10-01323-t003:** Comparison of patient characteristics between responders and nonresponders to HBV vaccination by univariate analysis at six months.

	Responders *n* = 96	Nonresponders *n* = 29	OR (95% CI)	*p*
Diabetes (*n*, %)	15 (16.5)	12 (35.3)	2.14 (1.11–4.09)	0.029
Alcohol etiology (*n*, %)	25 (27.5)	19 (55.9)	2.03 (1.30–3.18)	0.022
Liver cirrhosis (*n*, %)	29 (31.9)	25 (73.5)	2.30 (1.61–3.31)	<0.001
Stage F3–F4 by elastography (*n*, %)	30 (33.3)	23 (69.7)	2.09 (1.44–3.02)	<0.001
US signs of portal hypertension (*n*, %)	15 (16.5)	15 (44.1)	2.67 (1.47–4.86)	0.002

Abbreviations: CI: confidence interval; US: ultrasonographic.

**Table 4 vaccines-10-01323-t004:** Comparison of patient characteristics between responders and nonresponders to HBV vaccination by univariate analysis at twelve months.

	Responders *n* = 96	Nonresponders *n* = 29	OR (95% CI)	*p*
Alcohol etiology (*n*, %)	18 (24.3)	25 (51.0)	2.09 (1.29–3.39)	0.004
Liver cirrhosis (*n*, %)	19 (25.7)	33 (67.3)	2.67 (1.73–4.10)	<0.001
Stage F3-F4 by elastography (*n*, %)	20 (27.4)	31 (64.6)	2.40 (1.57–3.67)	<0.001
US signs of portal hypertension (*n*, %)	11 (14.9)	18 (36.7)	2.50 (1.30–4.80)	0.005

Abbreviations: CI: confidence interval; US: ultrasonographic.

## Data Availability

The data presented in this study are available on request from the corresponding author. The data are not publicly available due to privacy restrictions. All data that were analyzed during this study are included in this published article.

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
