# Peer review of "Efficacy of Hepatitis B Virus Vaccines HBVaxpro40© and Fendrix© in Patients with Chronic Liver Disease in Clinical Practice"

_vaccines, 2022, doi:10.3390/vaccines10081323_

Round 1

Reviewer 1 Report

Lines 44-45: "The protection offered by primary HBV vaccination with monovalent vaccine to healthy patients can persist for up to two decades."

Experts have recently recognized that such protection can probably persist life-long in most healthy subjects when immunized as children, despite the progressive loss of circulating antibodies below 10 U/mL. Actually you have made a similar comment in lines 291-293. I suggest you adapt your statement in lines 44-45. 

Lines 77-79: You propose to evaluate the effectiveness of the 2 vaccine regimens but you actually limit this assessment to antibody measures at 6 and 12 months. Attempts to measure other aspects of the immune response or long-term effectiveness /efficacy should be emphasized and this aspect of study limitation should be discussed. You could usefully expand on your interesting comment (lines 291-297 : "[after a booster dose in patients who lost anti-HBs over time,] a response rate was achieved in ... 87.5% (7/8) of ... losers of response"). Also, did you collect any other indirect evidence of efficacy; e.g. by diagnosing any later case of HBs infection in your study cohort?

Lines 100-101: Exclusion criterion mentioned as "or 6) nonimmunized HAV infection." Can you clarifiy what this means: previous anti-HAV vaccination? previous HAV infection in a non-HAV immunized individual? Other? Also the numbering should be 7) as exclusion criterion 6) is already given as pregnancy or breastfeeding.

Lines 113-115: "Patients were vaccinated with HBVAXPRO® 40 (0, 1 and 6 months) or FENDRIX® (0, 1, 2 and 6 months) depending on the type of vaccine available to the Health Department of the Catalan Government." Please clarify how and when either each group of the 2 vaccines were administered to patients, as this is quite a deviation from a classical random allocation of vaccines and may thus introduce some kind of bias in your study?

3.2 and 3.3 Results/ Vaccination efficacy: in addition to the response rates, it could also be interesting to mention the Geometric Mean Titres obtained with each vaccine at the several time points and if any significant difference was observed at any time point - or absence of difference? This could add some interesting information to your pooled mention about median anti-HBs levels in responders?

3.4 Adverse events: it is quite surprising that no AEs were reported during the study period, especially in such a high risk patient population! Can you clarify what AEs were solicited if any and what methodology was used?

Author Response

Lines 44-45: "The protection offered by primary HBV vaccination with monovalent vaccine to healthy patients can persist for up to two decades."

Experts have recently recognized that such protection can probably persist life-long in most healthy subjects when immunized as children, despite the progressive loss of circulating antibodies below 10 U/mL. Actually you have made a similar comment in lines 291-293. I suggest you adapt your statement in lines 44-45.

Changes made in lines 44-45.

Lines 77-79: You propose to evaluate the effectiveness of the 2 vaccine regimens but you actually limit this assessment to antibody measures at 6 and 12 months. Attempts to measure other aspects of the immune response or long-term effectiveness /efficacy should be emphasized and this aspect of study limitation should be discussed. You could usefully expand on your interesting comment (lines 291-297: "[after a booster dose in patients who lost anti-HBs over time,] a response rate was achieved in ... 87.5% (7/8) of ... losers of response"). Also, did you collect any other indirect evidence of efficacy; e.g. by diagnosing any later case of HBs infection in your study cohort?

Thank you for your interesting comment. Unfortunately, we did not measure other aspects of the immune response. As mentioned above, the vaccine probably remains effective even if protective antibodies are not detected in peripheral blood. But if they are negative, we cannot be sure that a patient will be protected. In studies carried out in the general population, patients with positive antiHBc have indeed been detected after vaccination, without evidence of symptomatic acute infection, nor evidence of chronic infection. This suggests that if a B virus infection contagion occurs in a previously vaccinated patient, the infection is mild and rarely chronic. We don't have enough follow-up of our patients to be able to answer this question.

Lines 100-101: Exclusion criterion mentioned as "or 6) nonimmunized HAV infection." Can you clarifiy what this means: previous anti-HAV vaccination? previous HAV infection in a non-HAV immunized individual? Other? Also the numbering should be 7) as exclusion criterion 6) is already given as pregnancy or breastfeeding.

Changes made in lines 100-101.

Lines 113-115: "Patients were vaccinated with HBVAXPRO® 40 (0, 1 and 6 months) or FENDRIX® (0, 1, 2 and 6 months) depending on the type of vaccine available to the Health Department of the Catalan Government." Please clarify how and when either each group of the 2 vaccines were administered to patients, as this is quite a deviation from a classical random allocation of vaccines and may thus introduce some kind of bias in your study?

Thank you for your excellent comment, we totally agree with you. This is one of the main limitations in our study as we have mentioned in the discussion (lines 300-303). As it is explained, the Health Department of the Catalan Government distributes the vaccines according to availability in the markets. Therefore, it is not our decision nor that of the Department of Health. For this reason, we mention this is a study performed in a clinical practice setting.  

3.2 and 3.3 Results/ Vaccination efficacy: in addition to the response rates, it could also be interesting to mention the Geometric Mean Titres obtained with each vaccine at the several time points and if any significant difference was observed at any time point - or absence of difference? This could add some interesting information to your pooled mention about median anti-HBs levels in responders?

We have calculated the Geometric Mean Titres at two, six and twelve months with each vaccine and have not found significant differences between the two vaccines at the different time points. We have added this information in the results section.

3.4 Adverse events: it is quite surprising that no AEs were reported during the study period, especially in such a high-risk patient population! Can you clarify what AEs were solicited if any and what methodology was used?

Our results regarding AEs are similar that that found in other clinical studies (S. Rodríguez-Tajes et al, J Viral Hepatol 2021; S.H. Engler et al, Eur J Gastroenterol Hepatol 2001; T. de Artaza et al, Gastroenterol Hepatol 2009; D. A Roni et al, Adv Virol 2013). The patients received the information related to the possible AEs described in the technical data sheet of both vaccines. All patients had the contact of our unit and were instructed to call/email us in case there was any AE In our study. There were not AEs that required any extra medical visit.

Reviewer 2 Report

An interesting study in which the results of vaccination with two vaccines are presented, one doubled adjuvanted (3-O-desacyl-4'-monophosphoryl lipid A and aluminium phosphate) and the other only aluminium ajuvanted in chronic patients. The study allows us to conclude that both vaccines were effective and safe, and can induce an adequate response without significant differences and highlights the importance of vaccination even in chronic patients.

Just a small suggestion: in the abstract and introduction mention that: The FENDRIX® vaccine is double adjuvant with adjuvant 3-O-desacyl-4'-monophosphoryl lipid A and aluminium phosphate and the HBVAXPRO® 40 is single adjuvant with aluminium hydroxyphosphate sulfate.

Author Response

Thank you for your suggestion. I add it in the abstract and introduction.